# Probing molecules in gas cells of subwavelength thickness with high frequency resolution

Guadalupe Garcia Arellano[1,2], Joao Carlos de Aquino Carvalho[1,2,3], Hippolyte Mouhanna[1,2], Esther Butery[1,2], Thierry Billeton[1,2], Frederic Du-Burck[1,2], Benoit Darquié [1,2], Isabelle Maurin[1,2] & Athanasios Laliotis [1,2] ✉

Miniaturizing and integrating atomic vapor cells is widely investigated for the purposes of fundamental measurements and technological applications such as quantum sensing. Extending such platforms to the realm of molecular physics is a fascinating prospect that paves the way for compact frequency metrology as well as for exploring light-matter interactions with complex quantum objects. Here, we perform molecular rovibrational spectroscopy in a thin-cell of micrometric thickness, comparable to excitation wavelengths. We operate the cell in two distinct regions of the electromagnetic spectrum, probing $v_1 + v_3$ resonances of acetylene at 1.530 μm, within the telecommunications wavelength range, as well as the $v_3$ and $v_2$ resonances of $SF_6$ and $NH_3$ respectively, in the mid-infrared fingerprint region around 10.55 μm. Thin-cell confinement allows linear sub-Doppler transmission spectroscopy due to the coherent Dicke narrowing effect, here demonstrated for molecular rovibrations. Our experiment can find applications extending to the fields of compact molecular frequency references, atmospheric physics or fundamental precision measurements.

Interfacing atomic and molecular gases with compact photonic platforms is a long-standing goal promising for quantum sensing[1–5], optical image processing[6], gas lasers and super-continuum sources[7,8], as well as time keeping[1] and frequency referencing applications[9]. Beyond their technological interest, compact hybrid platforms have an impact in fundamental studies of quantum electrodynamics in the presence of surfaces[10–12].

In the realm of atomic gases compact integrated atomic cells of mesoscopic size have been realized by microfabrication techniques[13] that find applications as portable atomic clocks[1] or magnetometers[2]. Beyond these works, subwavelength confinement in the micro and nanoregime has been studied in thin-cells[10,14,15], initially explored by Romer and Dicke[16] in the microwave regime. Such cells are now used for studying Casimir-Polder interactions[10,17], as well as cooperative[18]

and collective quantum effects[19], paving the way for a new generation of room temperature quantum technology devices. Finally, extreme 3D subwavelength miniaturization has been recently achieved by confining atoms in opals[20] or in nanostructured vapor cells[21].

In contrast to atoms, molecules have a plethora of rovibrations throughout the spectrum offering the possibility of ultra-narrow spectroscopy, with a linewidth essentially proportional to gas pressure. For this reason, molecular cells based on photonic platforms are expected to find important applications in frequency referencing, especially in the telecommunications spectral window, via nonlinear spectroscopy of weak, hard-to-saturate, acetylene inter-combination lines[22–24]. Hollow core fibers filled with molecular gas[9,23–25], tapered fiber devices[22] or even chip-based waveguides 'cladded' with a molecular gas[26] (inspired by their atomic analogues[27]) have been proposed

[1]Laboratoire de Physique des Lasers, Université Sorbonne Paris Nord, F-93430 Villetaneuse, France. [2]CNRS, UMR 7538, LPL, 99 Avenue J.-B. Clément, F-93430 Villetaneuse, France. [3]Departamento de Física, Universidade Federal de Pernambuco, Cidade Universitária, 50670-901 Recife, PE, Brasil. ✉e-mail: laliotis@univ-paris13.fr

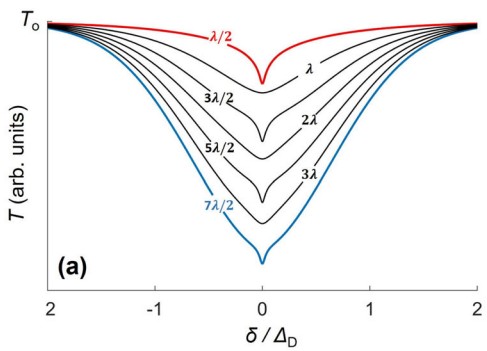

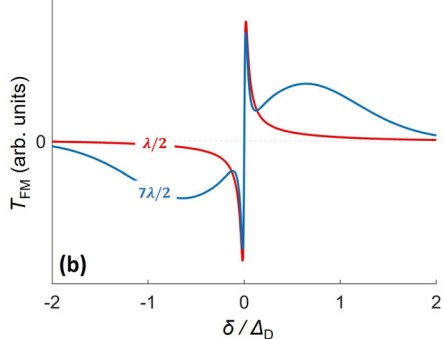

**Fig. 1 | Characteristics of thin-cell spectroscopy. a** Calculated thin-cell transmission ($T$) spectra as a function of the frequency detuning $\delta$ normalized by the Doppler width $\Delta_D = u_p / \lambda$ (where $u_p$ is the most probable velocity) for different cell thicknesses. The homogeneous linewidth is assumed to be $\Gamma = \Delta_D/30$. The transmission of an empty cell (without molecules) is denoted as $T_o$. The sub-Doppler contribution observed for $L = \lambda/2$ is revived when $L = (2n+1)\frac{\lambda}{2}$ (coherent Dicke-narrowing). This oscillatory behavior (the collapse and revival of the sub-Doppler peak) eventually disappears for larger L (when the collisional mean free path becomes smaller than the cell thickness) and a sub-Doppler peak is observed for all thicknesses. The Doppler contribution, strongly suppressed for $L = \lambda/2$, gradually builds up increasing with cell thickness, eventually overshadowing the sub-Doppler contributions for macrometric cells. To give an idea of the vertical scale, we note that thin-cell absorption is typically below ~$10^{-5}$ for our experiments. **b** Calculated FM-transmission spectrum ($T_{FM}$) for $L = \lambda/2$ (red curve) and $L = 7\lambda/2$ (blue curve), that are relevant for this work. The amplitude of the sub-Doppler (narrow) contribution is slightly smaller for $L = 7\lambda/2$. The dependence of the narrow peak amplitudes as a function of cell thickness is discussed in ref. 38. The curves are calculated following the models developed in[41,69] assuming that the reflectivity of both interfaces delimiting the thin-cell is zero. Here, we assumed that the $_{FM}$ signal ($T_{FM}$) is simply the frequency derivative of the direct transmission.

for this purpose. These devices confine molecules in two dimensions, but have a macroscopic interaction length.

Extending the sub-wavelength confinement studies performed on atomic vapors over the last 20 years to molecular gases has remained up to now very challenging. This is mainly because molecular rovibrations have small transition probabilities. Moreover, the strongest rovibrations fall in the mid-infrared, the molecular fingerprint region, where easy-to-use laser sources were not accessible before the advent of Quantum Cascade Lasers (QCL). Experiments have probed confined molecular gases[28,29] but this was achieved for high molecular densities compromising the frequency resolution[28,29]. So far, the only high-resolution (sub-Doppler) probing of molecular gases close to surfaces was achieved by selective reflection spectroscopy, where the effective optical confinement depends on the excitation wavelength[30]. These experiments could push Casimir-Polder studies further, unraveling fundamental effects linked to the complex molecular geometry[31,32] that are inaccessible with atoms.

Here, we probe low-pressure molecular gases confined in a thin-cell of micrometric thickness comparable to the excitation wavelength. We perform rovibrational spectroscopy of acetylene at 1.53 µm, as well as of $SF_6$ and $NH_3$ at 10.55 µm. For these wavelengths, the linear, low-power transmission through our cell presents sub-Doppler features, due to the coherent Dicke narrowing[14,16,33], with observed spectra that are very well interpreted by our theoretical models. The combination of linearity and high-resolution offers an original method of sub-Doppler molecular spectroscopy with significant advantages for enriching molecular databases and makes thin-cells attractive candidates for the fabrication of compact frequency references. We also discuss the potential of thin-cells for fundamental physics experiments with molecules.

## Results

### Coherent Dicke narrowing and cell thickness
Thin-cell spectroscopy under normal incidence has been studied extensively with dilute atomic vapors[14,15,33–38]. In thin-cells with thickness smaller than half the excitation wavelength ($\lambda/2$) the contribution of fast molecules (in the direction of the beam) is strongly inhibited due to transient broadening resulting from hard collisions with the cell walls. This leads to spectroscopic signals narrower than the Doppler profile. Nevertheless, linear thin-cell transmission spectroscopy also presents sub-Doppler features for larger cell thicknesses ($L$)[14,33]. These features, more pronounced when $L$ is an odd multiple of $\lambda/2$, are due to the additive contribution of all molecular velocities at $\delta = 0$, where $\delta$ is the laser frequency detuning[14,34,38]. The collapse and revival of the coherent Dicke narrowing, experimentally demonstrated in[14], is illustrated in Fig. 1a where the predicted linear thin-cell transmission spectra ($T$) is shown for different cell thicknesses. In Fig. 1b we show the frequency modulated (FM) thin-cell transmission signals for $L = \lambda/2$ and $7\lambda/2$, relevant for this work, for which the frequency resolution is mostly governed by the homogeneous transition linewidth ($\Gamma$).

### Thin-cell fabrication
The thin-cell fabricated for this experiment is shown in Fig. 2. The cell consists of two ZnSe windows separated by an annular spacer, made from commercial gold-foil of nominal 5 µm thickness. The windows are held together with screws (mechanical pressure). The gold spacer determines the cell thickness but also effectively seals the cell, acting like a vacuum flange due to strain hardening. A ~5 mm diameter hole is drilled into one of the windows allowing connection to a vacuum system via a metallic tube and a KF flange (Fig. 2). The above fabrication method does not require processing or optical contact of the windows, therefore providing great flexibility in the choice of dielectric. Here, we fabricated a uniform thickness ZnSe cell, which is a fragile material but has the advantage of being transparent throughout the near and mid-infrared.

After pumping, the cell thickness is measured by Fabry-Perot interferometry (see Methods) and ranges between 5.27–5.5 µm (Fig. 2c). In the most used area of the cell, the thickness is 5.35 µm ± 0.02 µm, a value that is close to $\lambda/2$ for the mid-infrared strong rovibrations of $SF_6$ and $NH_3$ molecules at 10.55 µm and to $7\lambda/2$ for the $C_2H_2$ overtones between 1.51–1.54 µm.

### Acetylene spectroscopy at telecommunication wavelengths
To demonstrate the potential of our device, we first probed the $\nu1 + \nu3$ P(9) transition of acetylene at 1.530 µm, which is the strongest acetylene rovibration in this wavelength range accessible with our laser system. Our thin-cell is typically filled with acetylene pressures ranging from 0.65–3 Torr, for which the homogeneous linewidth ($\Gamma$), determined by molecular collisions (see Table 1) remains significantly smaller than the Doppler width ($\Delta_D$ ~285 MHz).

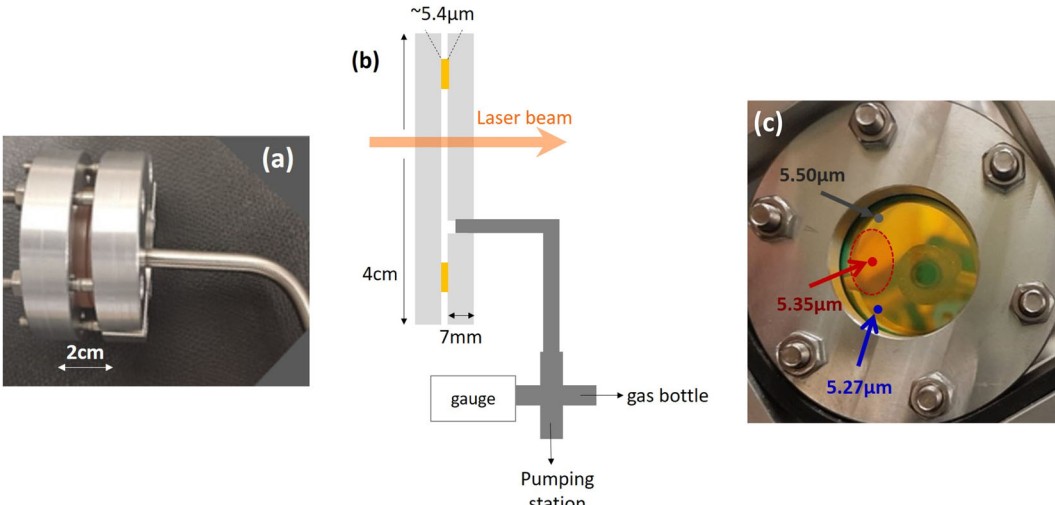

**Fig. 2 | The thin-cell fabricated for our experiments. a** Photograph of the thin-cell fabricated for our experiments. **b** Schematic of the thin-cell (not to scale). The resonant laser beam crosses the cell at normal incidence. One of the windows is attached to a vacuum tube that allows pumping and filling with molecular gas via electronically controlled valves (not shown here). The pressure is measured with a gauge inevitably positioned outside the cell. **c** Photograph of the thin-cell. This particular device was designed to have a relatively uniform thickness and therefore the windows were chosen to be ~7 mm thick, to minimize buckling under atmospheric pressure. In the central region of the cell (~1 cm², indicated with a red circle) the thickness is 5.35 μm ± 0.02 μm (corresponding to a parallelism of ~10 μrad between the two windows), while at the edges, the thickness can vary from 5.50 μm (grey point on the upper side) to 5.27 μm (blue point on the lower side).

**Table 1 | Relevant parameters for the molecular transitions probed in this work**

| Molecule | Line assignment | Absolute frequency (MHz) | Doppler width (MHz) | Observed pressure broadening (MHz/Torr) |
|---|---|---|---|---|
| $C_2H_2$ | P9 ($v_1 + v_3$) | 195 895 288.0 | 285 | 10.7 ± 1.8 |
| $SF_6$ | Q62E, Q62A2, Q62F2 ($v_3$) | 28 427 502.6 | 17 | 4.4 ± 0.4 |
| $NH_3$ | saP(1,0) ($v_2$) | 28 427 281.4 | 51 | 18 ± 1.7 |

The transition frequencies are taken from the HITRAN database[49]. The reported Full Width Half Maximum (FWHM) pressure (collisional) broadening is measured for molecular gases confined in thin cells for a pressure range of 0.65-3 Torr, 0.04-0.66 Torr, 0.075-0.3 Torr for $C_2H_2$, $SF_6$ and $NH_3$ gases respectively. The pressure measurements were performed with a Pirani sensor calibrated with a capacitive gauge the readings of which are independent of the nature of the gas. More details on pressure broadening and shift measurements are given in the Methods section.

For our experiment, we use an extended cavity diode whose frequency is scanned by applying a voltage to the piezoelectric actuator of the grating (see Methods). The FM amplitude and frequency are $M = 5$ MHz and $f_{FM} = 1$ kHz respectively. One part of the laser beam, with a power of ~2-3 mW, simply crosses the thin-cell at normal incidence and is subsequently detected by a Ge photodiode. We refer to the demodulated signal after lock-in detection as $T_{FM}$. An auxiliary saturated absorption experiment in a macroscopic (about 0.5 m length) cell provides a conventional narrow frequency reference in the volume. For this purpose, the laser beam is amplified by an Erbium Doped Fibre Amplifier (EDFA), providing ~200 mW with a beam size of ~1 mm, to saturate the weak acetylene rovibrations (see Methods).

Figure 3a shows thin-cell FM-transmission ($T_{FM}$) spectra for three acetylene pressures normalized by the transmission signal through the empty cell ($T_o$). A sub-Doppler structure remains visible for all curves. The spectra presented in Fig. 3, result from an overall averaging of ~200 individual ~3 min scans. We have succeeded in eliminating a parasitic signal baseline below the ~$10^{-8}$ level (smaller than statistical uncertainties) by subtracting consecutive scans with and without molecules (molecular pressure on/off modulation technique)[30]. For this, we used a system of electronically controlled valves that allow pumping and refilling of the cell at time scales on the order of 10–20 s (see Methods). The statistical noise level of this experiment is on the order of ~$2 \times 10^{-8}$, dominated by electronic detection noise. A larger signal-to-noise ratio could be achieved by using more laser power (while avoiding saturation effects) and by improvements on the set-up.

In Fig. 3a we plot the theoretically predicted spectra, fitted to the experimental data using the pressure broadened linewidth as a free parameter. Additionally, the theoretical curves are adjusted for amplitude and a small offset. Our fits allow us to extract the pressure broadening of the P(9) acetylene line which is measured to be 10.7 ± 1.8 MHz/Torr. This is consistent with values reported in the literature[22,23,39,40] (see also Methods) that can nevertheless vary between 8–13 MHz/Torr depending on the experiment.

The theoretical model assumes that collisions with the surface destroy laser-molecule coherence and the velocity of desorbed molecules follow a Maxwell-Boltzmann distribution. This allows us to calculate the transient molecular response for ballistic molecular trajectory from one wall to the other, with intermolecular collisions described using a pressure-dependent homogeneous linewidth. In addition to transient effects theoretical calculations account for Fabry-Perot cavity effects[41] and distortions due to FM modulation, calculated without any simplifying approximations[30,42]. The effects of laser linewidth (below 1 MHz) and of molecule-surface interactions can here be safely ignored[30]. Our models accurately reproduce the experimental spectra at the line center (sub-Doppler contribution) but also at the Doppler broadened wings.

Thin-cell theory is further tested in Fig. 3b, where the signal transmission, measured at different thicknesses (see Fig. 2c) for a pressure of 1.4 Torr, is almost perfectly superposed to the theoretically predicted spectra. The asymmetric lineshapes observed when the cell thickness deviates from $7\lambda/2$ are due to the mixing of the symmetric forward molecular response (transmission) with the asymmetric

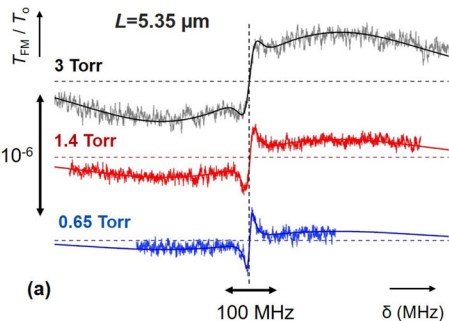

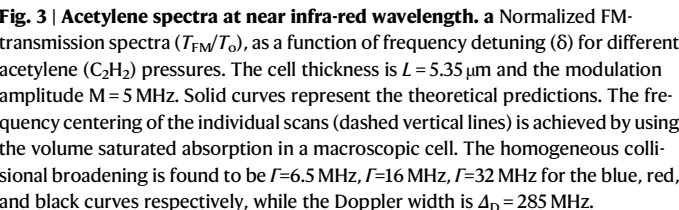

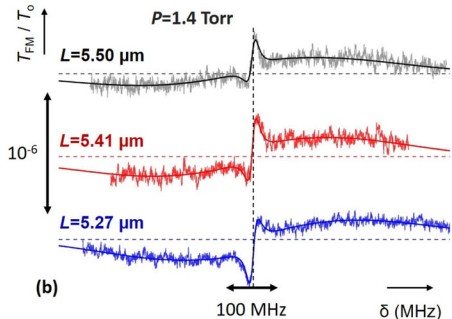

**Fig. 3 | Acetylene spectra at near infra-red wavelength. a** Normalized FM-transmission spectra ($T_{FM}/T_o$), as a function of frequency detuning ($\delta$) for different acetylene ($C_2H_2$) pressures. The cell thickness is $L = 5.35\,\mu m$ and the modulation amplitude M = 5 MHz. Solid curves represent the theoretical predictions. The frequency centering of the individual scans (dashed vertical lines) is achieved by using the volume saturated absorption in a macroscopic cell. The homogeneous collisional broadening is found to be $\Gamma$=6.5 MHz, $\Gamma$=16 MHz, $\Gamma$=32 MHz for the blue, red, and black curves respectively, while the Doppler width is $\Delta_D = 285$ MHz.

**b** Normalized FM-transmission spectra ($T_{FM}/T_o$) as a function of frequency detuning ($\delta$) for different cell thicknesses. The $C_2H_2$ pressure is 1.4 Torr and the modulation amplitude M = 5 MHz. The solid curves represent the predictions of the theoretical model for a fixed value of linewidth $\Gamma$=16 MHz. The asymmetries in the lineshape and the difference in amplitude, when cell thickness differs from its optimal $7\lambda/2$ value, are perfectly reproduced by the theory. Dashed horizontal lines represent the zero for each spectrum.

backward response (reflection) in the Fabry-Perot cavity of the cell. In Fig. 3b both linewidth and amplitude of the theoretical curves are fixed to the values extracted from Fig. 3a (1.4 Torr red curve) and only a small offset ($-10^{-8}$) is independently adjusted. The good parallelism between the cell windows, allows the beam to explore a uniform thickness and contributes to the remarkable agreement between experiment and theory.

## Mid-infrared rovibrational spectroscopy

We also studied the mid-infrared region of the spectrum where most molecular rovibrations can be found. In particular, the fingerprint region around 10 μm is of interest for precision measurements[43,44], and metrology[45], but also in the field of atmospheric physics and gas tracing[46–48]. We perform spectroscopy of $\nu_3$ and $\nu_2$ rovibrations of $SF_6$ and $NH_3$ molecules respectively (see Table 1), at 10.55 μm, both accessible with our laser set-up described in[30].

The QCL source used for this experiments emits ~5 mW of optical power at its output after optical isolation. The principles of our spectroscopic detection remain the same as those described for the $C_2H_2$ experiments. Here the FM modulation is applied on the laser current with a frequency $f_{FM} = 10$ kHz and an amplitude of M = 0.33 MHz. A saturated absorption spectrum is also simultaneously recorded for frequency calibration. QCL technology has not yet attained the maturity of telecom wavelength lasers. We have, therefore, developed a number of techniques to improve the QCL frequency scan and render it compatible with the needs of high-resolution spectroscopy[30].

Sulfur hexafluoride ($SF_6$), is a heavy spherical top molecule of octahedral geometry that presents a complex spectroscopic landscape. $SF_6$ is in the list of greenhouse gases and determination of both frequency and amplitude of its absorptions is required in order to measure the evolution of its atmospheric concentration[48]. The large transition probability of the $\nu_2$ rovibrations of $SF_6$, allows us to use lower pressures, achieving higher frequency resolution compared to acetylene spectroscopy. Ultimately the resolution of our mid-infrared experiments is limited by the QCL linewidth (~0.7 MHz FWHM).

Figure 4a shows the $SF_6$ thin-cell FM-transmission spectrum centered on the strong Q62E, Q62A2 and Q62F2 degenerate triplet. The adjacent, smaller transitions, probably corresponding to $SF_6$ hotbands, are not reported in the HITRAN database[49] but their frequency positions have been pinpointed in previous experiments using saturated absorption spectroscopy and a high precision wavemeter[30]. The curve of Fig. 4a results from averaging about 50 individual ~2 min scans, using the aforementioned molecular pressure on/off

modulation technique. The noise level is here ~$4 \times 10^{-8}$, slightly higher than the shot noise limit of ~$2 \times 10^{-8}$. The theoretical model, used to interpret the mid-infrared spectra also accounts for the influence of laser linewidth, which is comparable to the FM amplitude and the homogeneous pressure broadened linewidth. The fit allows us to extract the collisional broadening ($\Gamma$) and shift, as well as the relative transition amplitudes of the small and unidentified lines compared to the main strong triplet. Experiments at a pressure range of 0.04−0.66 mTorr give us a collisional broadening of $4.4 \pm 0.4$ MHz/Torr, while collisional shifts remain negligible.

Although these transition amplitudes were measured by selective reflection[30], thin-cells (here operated at $\lambda/2$ thickness) are simpler platforms for molecular spectroscopy providing a larger signal-to-noise ratio (see Methods). This experiment, allows us to measure amplitudes with an uncertainty below 5% with a better signal-to-noise ratio compared to[30] by roughly an order of magnitude. Our set-up could be used to access a wide range of $\nu_3$ rovibrations of $SF_6$ between 28.35-28.5 THz thus providing spectroscopic information that could be used to enrich molecular databases.

The ammonia ($NH_3$) molecule, a symmetric top of tetrahedral geometry, has a simple rovibrational spectrum with well-isolated transitions. For this reason, it is better suited for metrology or fundamental physics experiments. $NH_3$ has nevertheless a hyperfine structure with splittings as large as ~1 MHz (depending on the rovibrational level), that need to be taken into consideration when interpreting spectral lineshapes. Here, we probe the saP(1,0) transition, from the ground state to the first $\nu_2$ vibration, whose hyperfine structure is shown in the inset of Fig. 4b.

The thin-cell FM-transmission spectrum of $NH_3$ is shown in Fig. 4b for pressures of 31 mTorr and 94 mTorr. Further experiments (not shown here) were performed for pressures up to 300 mTorr providing the pressure broadening of $18 \pm 1.7$ MHz/Torr (Table 1). The scans are recorded using the same parameters and conditions as in the case of $SF_6$ (Fig. 4a). As pressure decreases, the hyperfine structure of the saP(1,0) rovibration becomes more resolved and apparent on the spectra. Reducing the pressure below 30 mTorr does not significantly improve the resolution (limited by laser linewidth) but simply reduces the signal amplitude. The solid lines represent the theoretical fits, where both positions and relative amplitudes of the hyperfine transitions are fixed to their theoretically expected values[50]. Lineshape analysis allows us to measure a collisional shift of $3.7 \pm 1$ MHz/Torr for the saP(1,0) $\nu_2$ transition of $NH_3$. We observe no differential pressure shift between the hyperfine components of the examined transition,

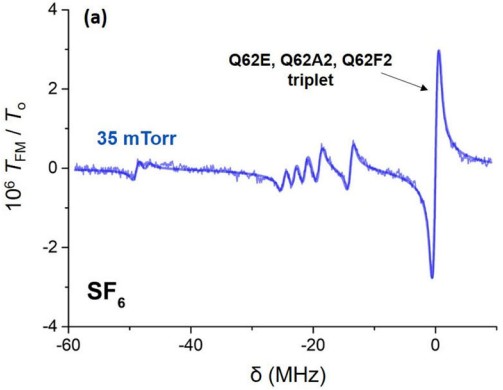

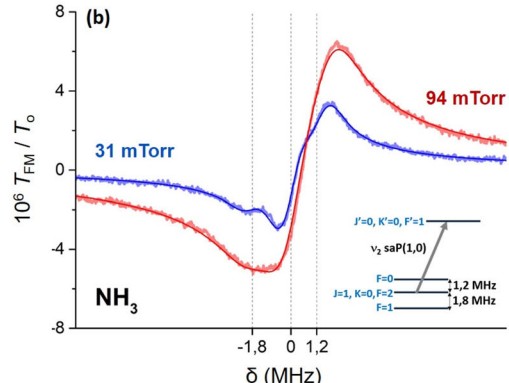

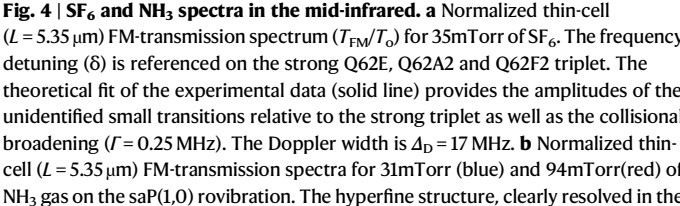

**Fig. 4 | SF$_6$ and NH$_3$ spectra in the mid-infrared. a** Normalized thin-cell ($L$ = 5.35 μm) FM-transmission spectrum ($T_{FM}/T_o$) for 35mTorr of SF$_6$. The frequency detuning ($\delta$) is referenced on the strong Q62E, Q62A2 and Q62F2 triplet. The theoretical fit of the experimental data (solid line) provides the amplitudes of the unidentified small transitions relative to the strong triplet as well as the collisional broadening ($\Gamma$ = 0.25 MHz). The Doppler width is $\Delta_D$ = 17 MHz. **b** Normalized thin-cell ($L$ = 5.35 μm) FM-transmission spectra for 31mTorr (blue) and 94mTorr(red) of NH$_3$ gas on the saP(1,0) rovibration. The hyperfine structure, clearly resolved in the 31mTorr (blue) spectrum, is shown at the inset and the positions of the hyperfine transitions are marked with vertical dashed lines on the NH$_3$ spectrum. The relative amplitude between the hyperfine components is fixed to its theoretical value (1:5:3) for the ($F = 0 \rightarrow F' = 1$, $F = 2 \rightarrow F' = 1$ and $F = 1 \rightarrow F' = 1$ transitions respectively. The extracted collisional broadenings are $\Gamma$=0.55 MHz and $\Gamma$ = 1.7 MHz for 31mTorr and 94mTorr respectively, while the Doppler with is $\Delta_D$ = 51 MHz. In all fits in (**a**) and (**b**) the laser linewidth is considered to have a Gaussian profile with a 0.7 MHz FWHM. The FM amplitude is $M$ = 0.33 MHz.

while at low pressures the frequency spacing between the hyperfine components is consistent, to within 0.1 MHz, with previously reported values (shown on the inset of Fig. 4b).

## Discussion

We probed molecular gases in a thin cell of micrometric thickness corresponding to $\lambda/2$ for SF$_6$ and NH$_3$ rovibrations at 10.55 μm and 7$\lambda$/2 for acetylene transitions at 1.53 μm. The coherent Dicke narrowing[16], demonstrated for rovibrational transitions, allows us to obtain high-resolution, sub-Doppler transmission signals without resorting to nonlinear spectroscopic schemes. The thin-cell platforms presented here can be used to probe many molecular transitions spanning through a wide range of the electromagnetic spectrum.

One of our major results is the measurement of transmission signal amplitudes and lineshapes of subwavelength confined molecules that are very well reproduced by theory. This remarkable observation suggests that our experiment is compatible with the assumptions included in the models. The most important question concerns molecule-surface collisions and in particular their effect on the molecule-laser interaction[28,51] as well as the velocity distribution[52–54] and the partition function (redistribution in rotational states) of the molecules departing from the surface (desorbed molecules)[55]. The common assumption that collisions with the surface destroy the laser-particle coherence is justified[28], but the case of rovibrational spectroscopy needs further investigations since molecule–surface potentials can be very similar between the two probed states[51]. In contrast to atoms, molecules offer the possibility of ultra-high resolution spectroscopy thus allowing for more strenuous tests of the Maxwell-Boltzmann distribution. Additionally, studies of the signal amplitude of different rovibrations inside thin-cells can offer information on the molecular partition function of confined gases. Molecular thin-cells are therefore excellent testbeds for exploring the thermodynamics of confined gases.

A fascinating perspective is to strongly confine molecular gases in the nanometric regime[10] and perform spectroscopy of the Casimir-Polder interaction with molecules. Molecule-surface interactions present a multidisciplinary interest (from QED theory to physical chemistry applications) but experimental investigations remain scarce[56,57]. The fabrication process presented here is particularly suited for Casimir-Polder studies, as it offers high flexibility in the use of dielectric windows. In contrast to atoms, molecules could be probed in thin-cells operated below room temperatures, which can be of interest for studies of

thermal Casimir-Polder effects[55,58,59]. Molecular cells can also be operated under extreme molecular densities, which could prove an advantage for probing cooperative[18] or collective effects such as superradiance[60].

Our experiment provides an alternative technique for rovibrational molecular spectroscopy that combines linearity and crossover-transition-free sub-Doppler resolution. This allows simultaneous measurement of both transition amplitudes and frequencies with a significantly better signal amplitude than selective reflection. Thin-cell spectroscopy can therefore contribute to the compilation of molecular databases, essential for retrieving atmospheric concentrations and interpreting astrophysical spectra, complementing traditional methods such as Fourier transform infrared (FTIR) spectroscopy or Doppler broadened absorption spectroscopy that suffers from low-frequency resolution, or saturated absorption whose inherent non-linearity makes difficult the extraction of transition amplitudes. The advantages of using thin-cells are more evident for heavy atmospheric species such as SF$_6$ with low-lying vibrational modes, which exhibit dense rotational structures and many hot-bands, impossible to resolve by FTIR or Doppler spectroscopy, and for which the molecular databases remain largely incomplete[61].

The subject of SF$_6$ hot-bands is an open question in molecular spectroscopy, many of which still remain largely neglected in the HITRAN database even after a recent update[49]. Including thin-cell spectroscopic data in a global analysis of SF$_6$ spectra (such as the one performed in[61]) could potentially offer unprecedented information on hot-band transition amplitudes and positions for SF$_6$ and other heavy atmospheric molecules (ClONO$_2$, CF$_4$) in the future. Additionally, thin-cell spectroscopy can constitute a stringent test for validating the most advanced ab initio quantum mechanical calculations of the complete band structure of SF$_6$[62,63] and other complex molecules.

Dicke-narrowed molecular thin-cell spectroscopy is an important development in the field of compact frequency references at telecommunication wavelengths allowing high frequency-resolution with a simple, one-beam, versatile, low-power set-up, in contrast to saturation spectroscopy. Gaining a factor of 100 in signal amplitude could allow stabilization of a telecommunications laser on the thin-cell transmission, whose stability can subsequently be compared against a voluminous frequency reference obtained in a macroscopic cell[64]. Realizing a resonant high-finesse Fabry-Perot cavity could be a viable route towards signal amplification by a factor roughly proportional to

the finesse (see Methods). This could be achieved by depositing dielectric mirrors of modest reflectivity (~98% for an amplification of 100, with a cavity finesse on the order of 150) on the cell windows. Alternatively, fabricating a stack of cells (successively piling windows separated by spacers) or using a multi-pass technique could also be explored. Finally, micro-fabricated ultra-compact thin-cells, as reported in[13,65] for atomic vapor cells, could also be an interesting prospect for acetylene spectroscopy.

## Methods

### Gas cells, thickness and pressure measurements

The micrometric thin-cell consists of 2 ZnSe windows, with a wedge of about $2^{\circ}$, separated by an annular spacer. For the purposes of this experiment, we used a Goodfellow gold-foil as a spacer with nominal thickness of $5\,\mu m$ but we have also tested sputter coating and evaporation techniques (using a mask) that allow us to achieve a very good control of the spacer thickness. When applying mechanical pressure to seal the cell, the two external interfaces are aligned to be parallel to each other thus avoiding stress that could fracture the windows. The internal faces of the windows (the actual thin-cell walls) are uncoated, while the external interfaces are anti-reflection (AR) coated to avoid, as much as possible, parasitic reflections. The coating is optimal at $10.6\,\mu m$ ($NH_3$ and $SF_6$ experiments) giving a reflectivity of about 1%, but at $1.53\,\mu m$ ($C_2H_2$ experiments) its reflectivity is about 10%. The reflectivity of an uncoated ZnSe surface is ~17% and 18% for $10.6\,\mu m$ and $1.53\,\mu m$ respectively, creating a small finesse optical cavity. The curvature of the internal window interfaces (and possible deposited contaminants) can introduce variations (and uncertainties) to the cell thickness.

The thickness of the thin cells has been mapped by Fabry-Perot interferometry with an uncertainty of a few nm using three different lasers: a He-Ne at 633 nm, a diode laser at 852 nm, and the extended cavity laser emitting from 1519 nm to 1540 nm[10]. The three beams are aligned and are used to measure the thin cell reflectivity at different wavelengths and different spots of the cell. The thickness gradient is here small, allowing us to use a relatively big beam size of 1 mm diameter.

The macroscopic cells used to obtain an auxiliary saturated absorption reference are metal tubes sealed by windows glued at the ends. For the $C_2H_2$ experiment, we use a 45 cm long cell sealed by glass windows whereas for the $SF_6$ and $NH_3$ experiments we use a 15 cm long cell sealed by ZnSe windows (AR coated at both sides).

The thin-cell is connected to two motorized valves and a pressure gauge. We use a valve "*Pfeiffer Vacuum GmbH RME series*" to control the gas flow into the cell, and a valve "*Pfeiffer Vacuum GmbH EVR 116*" to control the connection to the pumping station. Both valves are actuated with an external computer-controlled voltage.

To determine the pressure inside the cells we use dual gauges (Pirani and cold cathode gauge) with a range extending from $10^{-6}$ mTorr up to almost atmospheric pressure. The Pirani gauge, which operates in the pressure range where our experiments are performed ($\approx$20-4000 mTorr) for $C_2H_2$, $SF_6$, or $NH_3$ gases, was later calibrated by a capacitive gauge (readings independent of gas type). This eliminates an important source of systematic errors in the pressure measurement and gives us more confidence in the reported pressure broadenings and shifts. Gauge calibration can be a source of discrepancies between our measurements and the values of pressure broadening reported in the literature.

### Frequency scan of the lasers

The extended cavity laser emitting around 1530 nm, used for $C_2H_2$ spectroscopy, is scanned by applying a voltage to the piezoelectric actuator of the grating. The laser detuning is calibrated with an uncertainty of ~1%, by using a BRISTOL 771B-MIR (1–12 $\mu m$) wavemeter and this measurement is subsequently corroborated by fitting the

linear Doppler absorption spectrum of acetylene at room temperature with a Voigt profile. Saturated absorption in the volume gives an absolute molecular reference with which the frequency scans are referenced with an accuracy better than the MHz.

The frequency drift of the QCL laser emitting around $10.55\,\mu m$ is significantly more important and the frequency scans are achieved by loosely frequency stabilizing the laser to a linear absorption profile while applying an offset voltage to the error signal. The technique is described in detail in previous works[30].

A modulation on the laser frequency ($f$), is applied to both lasers, $f(t) = f_o + M \sin(2\pi f_{FM} t)$, where $f_o$ is the central frequency, $M$ is the modulation amplitude and $f_{FM}$ is the modulation frequency. The frequency modulation (FM) is applied on the current of the $10.55\,\mu m$ emitting QCL and on the piezoelectric actuator of the 1530 nm laser.

### Saturated absorption spectroscopy

The $C_2H_2$ inter-combination lines in the telecommunications windows are relatively weak, with a dipole moment matrix element of $\mu = 3.6 \times 10^{-32}$ Cm[22], in contrast to the much stronger $NH_3$ and $SF_6$ rovibrations with $\mu = 1.4 \times 10^{-30}$ Cm[66] and $\mu = 0.8 \times 10^{-30}$ Cm[67] respectively. Therefore, the saturation intensity $I_{sat}$ ($I_{sat} \propto \Gamma^2 / \mu^2$, where $\Gamma$ is the homogeneous linewidth) is significantly higher for $C_2H_2$ ($I_{sat}$ ~ 50 W/cm²) than for $NH_3$ ($I_{sat}$ ~ 30 mW/cm²) or $SF_6$ ($I_{sat}$ ~ 80 mW/cm²) rovibrations. Here the values of saturation intensity are indicatively given for a linewidth of 1 MHz.

In order to saturate the acetylene transitions, the laser beam is amplified with an EDFA which provides 23.5 dBm (~200 mW) at its output. In the experimental set-up we use one single beam (pump beam) which is retro-reflected and attenuated, before passing through the cell, to serve as the probe beam. The pump beam size is about 1 mm (pump intensity ~20 W/cm²).

In the case of $NH_3$ and $SF_6$, saturated absorption spectroscopy is performed on a 15 cm cell using a similar set-up (here the source is a QCL source). Here, the amplification step is not necessary and saturation can be observed with a maximum optical power of ~1.5 mW and a beam diameter of about 2 mm.

The saturated absorption spectra can also be demodulated at the FM frequency allowing us to observe the derivative of the saturated absorption signal and obtain a frequency reference.

### Background elimination: molecular pressure on/off modulation technique

The molecular thin-cell transmission spectra are always 'contaminated' by a parasitic background resulting probably from the interference between the transmitted laser beam and parasitic reflections. Due to slow thermal fluctuations or small mechanical movements, the interferometric background slowly drifts, making its shape difficult to predict. To eliminate this parasitic signal, we have implemented the on/off pressure modulation technique mentioned in the main text and explained in[30]. Here, we recall the main steps of this technique.

1. Pumping the cell: First, the thin-cell is pumped by opening the aperture of the motorized valve which connects it to the vacuum pump. The valve is closed after reaching a pressure $P_{vaccum}$ of around 0.1mTorr and a frequency scan A (without molecules) is recorded.
2. Filling the cell with molecular gas: By opening the aperture of the solenoid valve, the cell is filled with molecular gas up to a pressure $P_{gas}$. After reaching the desired pressure ($P_{gas}$) the valve is closed and a frequency scan B (with molecules) is recorded.
3. Pumping the cell: The cell is pumped again and a frequency scan C (without molecules) is recorded.

The above steps are fully automated using computer control. Pumping and filling the cells takes 10–20 seconds, while scans A, B, and C are recorded within 120–180 seconds time depending on the

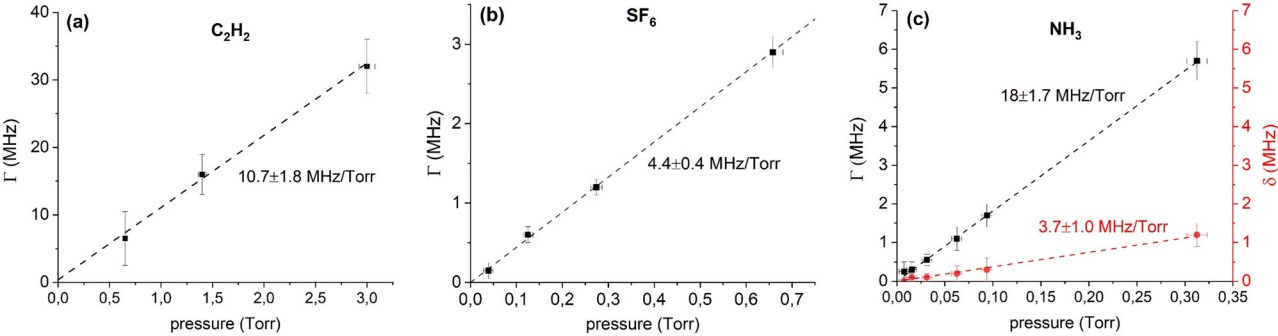

**Fig. 5 | Observed collisional broadenings and shifts.** Homogeneous linewidth Γ (black squares) and collisional shift δ (red circles, if applicable) as measured in our experiment for the P9 v1 + v3 rovibration of $C_2H_2$ (**a**), v3 rovibrations of $SF_6$ (**b**) and P(0,1) rovibration of $NH_3$ (**c**). The error bars on Γ and δ are conservatively extracted from lineshape analysis while the error bars on pressure include statistical fluctuations (5-10mTorr) of the readings and systematic uncertainties (3%) due to pressure gauge calibration.

required frequency resolution. After detection, the frequency scale of all scans is calibrated using the saturated absorption reference. The difference between scan B and the average of scans A and C is the 'pure' molecular thin-cell transmission. The cycle is then repeated N times (typically 20–200 times), allowing us to reduce the statistical noise typically below $10^{-7}$ (depending on the experiments and number of scans) and the residual interferometric background down to the $10^{-8}$ level for $C_2H_2$ experiments and down to the $10^{-7}$ level for $NH_3$ and $SF_6$ experiments.

### Fitting
The theoretical thin-cell transmission spectra are calculated using the theory developed in[41] ignoring Casimir-Polder interactions and assuming a Maxwell-Boltzmann distribution of molecular velocities inside the cell. After the theoretical 'direct' transmission curves have been calculated, we account for FM demodulation and the effects of the finite laser linewidth by convolving with an assumed Gaussian frequency distribution. The FWHM of the Gaussian function, that represents the laser linewidth, is fixed by independent measurements. For the mid-infrared experiments, the laser linewidth was experimentally estimated to be ~0.7 MHz (this is slightly higher than the value 0.6 MHz reported in ref. [30], probably because of small differences in the laser frequency stabilization loop). For the near-infrared experiments we have experimentally verified that the laser linewidth is well below 0.5 MHz and it can therefore safely be ignored in the fits.

To fit the spectra, we first produce a theoretical spectrum using an initial estimate of the value of Γ (collisional broadening). The curve is then adjusted for amplitude, offset and a small frequency shift to fit the experimental spectrum. We iterate the process for different values of Γ, until the best fit (the value that minimizes the least square difference) is identified. This gives us the value of collisional broadening, transition amplitude, and collisional shift.

### Thin-cell transmission and selective reflection spectroscopy
For simplicity, we discuss a symmetric cell made from two interfaces with a reflection coefficient $r$. In the infinite Doppler approximation $\Delta_D \gg \Gamma$ and assuming that $M \ll \Gamma$, the normalized FM transmission $T_{FM}/T_o$ through a cell of $\frac{\lambda}{2}$ thickness is given by[41]:

$$\frac{T_{FM}}{T_o} = 8 A M N \mu^2 \frac{\lambda}{u_p} \frac{(1+r)^2}{1-r^2} \frac{\delta}{\left(\frac{\Gamma}{2}\right)^2 + \delta^2} \qquad (1)$$

where $N$ is the molecular density, $\mu$ the transition dipole moment matrix element, $u_p$ is the most probable velocity and $A$ a constant.

The normalized FM selective reflection signal (frequency modulated resonant reflection from an infinitely long cell at normal incidence assuming an infinitely small but not strictly null absorption),

calculated with the same formalism, is:

$$\frac{S_{SRFM}}{S_R^o} = - 2 A M N \mu^2 \frac{\lambda}{u_p} \frac{(1-r^2)}{r} \frac{\delta}{\left(\frac{\Gamma}{2}\right)^2 + \delta^2} \qquad (2)$$

where $S_R^o = r^2$ is the reflected signal in the absence of molecules.

To compare the normalized FM thin-cell transmission signal reported here (for a thin cell of $\lambda/2$) and the normalized FM selective reflection signal of[30], we use the reflection coefficient ($r = 0.42$) of our ZnSe windows and account for the different modulation amplitudes: 0.33 MHz and 0.25 MHz for thin-cell and selective reflection spectroscopy respectively. We find that thin cell spectroscopy is expected to provide a larger amplitude by a factor of ~6.6 and a larger signal-to-noise ratio by ~11.9 (assuming the same incident power and a shot-noise limited experiment) compared to selective reflection. In our experiments, the observed amplitude ratio is slightly larger, roughly between 7 and 9, depending on the type of gas ($SF_6$ or $NH_3$) and the pressure. This small difference is probably due to the fact that in ref. [30] some mirrors were vibrated to reduce the parasitic background. This technique was later found to slightly degrade the amplitude and noise of our signals and was not used for thin cell spectroscopy.

### Pressure broadening and shift
Following the above process for thin-cell experiments we have found that the collisional broadening is $10.7 \pm 1.8$ MHz/Torr, $4.4 \pm 0.4$ MHz/Torr and $18 \pm 1.7$ MHz/Torr and for the $C_2H_2$, $SF_6$ and $NH_3$ rovibrations respectively. The collisional shift (red line in Fig. 5c) is measured to be $+3.7 \pm 1$ MHz/Torr in the case of $NH_3$ but stays negligible in the case of $SF_6$ and $C_2H_2$ (< 0.2 MHz for the explored pressures). In Fig. 5 we show the homogeneous linewidth (Γ) and shift (δ), extracted from the fits of the thin-cell transmission lineshapes as a function of pressure. The error bars on Γ and δ are conservatively extracted from our lineshape analysis. The pressure was measured with Pirani gauges that were subsequently calibrated with a capacitive gauge, whose readings are independent of the nature of the gas. The error bars on the pressure readings are due to statistical fluctuations of ~5-10mTorr and systematic errors of ~3% due to the calibration process.

In these experiments, the pressure gauge is naturally placed outside the thin-cell (see Fig. 2a of the main text). Thus a pressure differential cannot be excluded (cell thickness is ~5 μm), contrary to selective reflection measurements[30], performed with the same pressure gauge (and the same set-up) but in a macroscopic cell. Nevertheless, both techniques yield comparable pressure broadenings suggesting that gas pressure readings outside the cell also reflect the pressure inside the cell. Using selective reflection, we found 4 MHz/Torr and 21 MHz/Torr broadening for the same $SF_6$ and $NH_3$ rovibrations respectively with an

error bar on the order of 10% (here we have corrected the pressure readings reported in[30] assuming the same calibration factors).

In the case of $C_2H_2$ (no selective reflection measurements available) we have performed independent saturated absorption experiments for different gas pressures ranging from 20mTorr up to 500mTorr. For these dedicated measurements, we have separated the pump (~200 mW after amplification) and probe (~3 mW) beams and have applied an amplitude modulation on the pump, which allows us to eliminate the Doppler broadened background. At low pressures (below 100mTorr) the saturated absorption profile is well fitted by a Lorentzian function. The minimum achievable linewidth is ~2.5 MHz, probably limited by transit time broadening, small angle between pump and probe beams or laser linewidth. At higher pressures (above 100mTorr) the narrow saturated absorption (still interpreted by a Lorentzian profile) sits on a broader pedestal, probably due to velocity-changing collisions, that becomes more prominent as pressure increases. Eventually the narrow saturated absorption disappears for pressures above 500mTorr. The pressure broadening, extracted purely by analysis of the narrow component, is measured to be ~15 MHz/Torr slightly higher than the values obtained in the thin cell. The discrepancy between the two experiments can be due to velocity-changing collisions that can give rise to a nonlinear dependence of linewidth on pressure (see[68] and references therein) and whose effects could be different between linear (thin-cell) and nonlinear (saturated absorption) spectroscopic schemes.

## Data availability
The experimental data generated in this study are deposited in the zenodo database and are available under the accession code https://zenodo.org/records/10499906. Further data can be provided by the corresponding author upon request.

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

## Acknowledgements

We acknowledge financial support from the ANR project SQUAT (Grant No. ANR-20-CE92-0006-01), the LABEX Cluster of Excellence FIRST-TF (ANR-10-LABX-48-01) and the PN-GRAM. We thank Daniel Bloch for discussions throughout the duration of the project and comments on the text. We also thank Jean-Michel Hartmann, Martial Ducloy, Livio Gianfrani for interesting exchanges and Yannis Pargoire for his contribution in the automation of the molecular pressure on/off modulation technique.

## Author contributions

The experiment was built by G.G.A., J.C.A.C., I.M., E.B., A.L. The data was taken by G.G.A., J.C.A.C., E.B., H.M. The fits were performed by G.G.A., I.M., H.M., A.L. The cell was fabricated by T.B., I.M., A.L. Data acquisition and automation via computer was managed by I.M with the participation of H.M. B.D. has provided expertise in molecular spectroscopy and participated in data interpretation in discussions with A.L. F.dB. has provided expertise in frequency metrology and participated in data interpretation in discussions with A.L. The manuscript and the reply to referee's comments were written by A.L. with the participation of B.D. after discussions with F.dB., I.M., G.G.A., J.C.A.C. The project was coordinated by A.L.

## Competing interests

The authors declare no competing interests.
