## [Peer Review File · Nature Communications]

Probing molecules in gas cells of subwavelength thickness with high frequency resolutionREVIEWER COMMENTS

Reviewer #1 (Remarks to the Author):

Report on the manuscript entitled "Probing molecules in gas cells of subwavelength thickness with high frequency resolution" by G. Garcia Arellano et al., submitted to Nature Communications.

General comment.

This paper describes the observation of linear sub-Doppler molecular features in gas cells having sub-wavelength thickness, allowing for an effective coherent Dicke narrowing and Doppler broadening suppression. According to the best of my knowledge this is the first observation of such signals on molecules, being this topic the subject of studies exclusively performed on atomic samples. Moreover, the work has been performed in two different spectral regions, namely in the telecommunications range and in the MIR region, and on three different molecule (acetylene, ammonia and SF₆), showing the applicability of the methodology to any molecule. On the other hand, as clearly pointed out by the authors, this topic is of great interest for a quite wide scientific community, spanning from the frequency metrology to the quantum sensing world. Definitely, these are very important arguments that make the paper suitable for publication. In addition, the elements of novelty, in principle, could deserve publication on a very impact journal.

The text is clear, previous works are properly cited and the experimental results (in particular those described in Figures 3 and 4) are impressive in terms of signal-to-noise ratio, especially if compared to the elusiveness of the signals the authors are looking for.

However, there are several major issues that, in my opinion, should be addressed and considered by the authors before publication on Nature Communications (NC).

Major concerns:

- 1) There is no mention to an absolute frequency determination and the quality of the calibration of the frequency axis seems to be quite poor (1% in relative term) as compared to the state of the art. Moreover, in consideration of a possible application of the method for frequency metrology, I believe that some extra efforts should be done in this direction. Why do not to use a frequency comb as reference (at least in the 1.5 micron region)?
- 2) What about uncertainties? What about possible systematics effects? An overall uncertainty budget has been completely omitted. From the "statistical" point of view, it is surprising for me that the observed pressure broadening coefficients are given without uncertainty. Are they derived from repeated measurements? Are they experimental results? How can the authors claim consistency with literature values?

3) Results. From a quantitative point of view, the experiment provides results that are limited to pressure broadening coefficients (of one transition of each investigated molecule). This is not too much. Maybe, it is enough for a proof of principle experiment, but not for the initial ambitions of the work.

4) Laser sources used in the experiment. It seems to me that they represent the predominant limitation for the experiment (at least in the MIR. In the case of acetylene, pressure broadening conceals this limitation). This is disappointing. I am sure that the authors (see *Optica* 6 411-423, 2019) could have access to a "more powerful" and SI traceable QC laser, not limited to a linewidth at the MHz level. Again, as a proof of principle experiment this is not a problem, but when the publication targets for NC, extra efforts have to be done.

5) The authors have in their hands a very powerful method. Indeed, the hyperfine structure of ammonia has been observed. Why do not provide evidence of consistency (at least in terms of detuning of the different components) with previous experimental results or theoretical data?

6) "Discussion" paragraph. Usually, it should be a physical insight related to the results of the experiment described in the manuscript. In this work, it is a description of what can be done in future, as prospective. It is absolutely not quantitative.

7) "Thin-cell fabrication" paragraph: More appropriate for a technical report (or for the Method section).

Reviewer #2 (Remarks to the Author):

I enjoyed reading the paper by Garcia Arellano et al.. The paper describes performing vibrational spectroscopy on molecules that are confined in a cell that is comparable to the size of the wavelength of the light used to excite the molecules. As a result of the hindered motion of the molecules in the direction of the laser beam, the measured Doppler profile is modified markedly, and displays features that are more narrow than the Doppler distribution at the ambient temperature. The paper presents measurements on this effect in vibrational transitions in molecules in the infrared for the first time. The authors were able to build a thin cell with a very uniform spacing – tested using an interferometric technique, and used an elegant technique to reduce parasitic background. As a result, the lines are measured with a very high signal to noise ratio, providing textbook examples of the coherent Dicke effect. In my opinion this warrants publication of these results in *Nature Communications*.

It's hard for me to judge how realistic the proposed applications of this technique are. The obtained resolution and signal-to-noise-ratio, which are coupled as a lower pressure will lead to a higher resolution and a lower signal-to-noise ratio, should be increased by at least a factor of 100 before becoming comparable to state-of-the-art saturation spectroscopy. The authors suggest that a resonant high-finesse Fabry-Perot achieved by depositing dielectric coatings on the inner side of the cell windows could be a way to achieve this. It would be useful if it would be mentioned what the Q-factor of such a cavity would be. Is the currently achieved alignment sufficient for effective enhancement? Also it would be helpful to mention how the particular transitions were picked – I assume they are simple the one closest to the wavelength that fits the cell!?

In conclusion this is a well written paper on a beautiful and important experiment that is thoroughly analyzed and discussed. In my opinion this should be published in Nature Communications after the authors have considered the two points above.

Reviewer #3 (Remarks to the Author):

The authors show a nice application of thin-cell spectroscopy to molecular physics. They present data showing sub-Doppler resolution of the order of tens to hundreds of MHz in two different spectral ranges, telecommunications and 10 μm , for three different molecules. The results are particularly relevant for the potential to implement frequency references in the mid IR and high-resolution spectroscopy of species with congested spectra. I recommend this manuscript for publication in Nature Communications.

I have only minor remarks.

1. In the introduction, the authors correctly cite the relevant literature and note that thin-cell molecular spectroscopy has been elusive until now. I find difficult to guess why the experiments presented here have not been done years ago, for all necessary ingredients seem to be available now for quite some time. I would appreciate if the authors could make an effort to rationalise such delay. What are the key points that allowed the present measurements? As a reader I would find this interesting.
2. Caption to Fig 1: I would repeat at the text beginning that the data are calculations (is stated only in the main text).
3. Caption to Fig 1: I find this sentence strange: "This oscillatory, coherent, behavior eventually disappears for larger L (when the collisional mean free path becomes smaller than the cell thickness) and a sub-Doppler peak is observed for all thicknesses". First they say that the oscillatory behaviour eventually disappears and then they say that is observed for all thicknesses. I would reformulate this and the previous sentences as: "The sub-Doppler contribution observed for $L=\lambda/2$ is revived when $L=(2n+1)\lambda$ (coherent Dicke-narrowing). It is observed up to $7\lambda/2$ and disappears for larger L (when the collisional mean free path becomes smaller than the cell thickness)". Or something similar.
4. Caption to Fig 1: Since they give an idea of the vertical scale, would it be possible to write a non-arbitrary scale to the vertical axes? Perhaps in terms of some relevant parameters used in the simulations?
5. In the discussion "Gaining a few orders of magnitude in signal can allow stabilization of a telecommunications laser...". Can the authors give a more precise estimate of the required gain and quantify the expected gain from the proposed improvements? How far away are those applications?

6. Discussion: Something is missing from the sentence: “A global analysis of high resolution SF₆ spectra (such as the one performed in [54]), including thin-cell spectroscopy experiments could potentially offer unprecedented information on hot-band transition amplitudes and positions for SF₆ and other heavy atmospheric molecules...”. Perhaps an initial “In”?

7. Discussion: “Furthermore, the flexibility in temperature operation, coupled with high-frequency resolution and high velocity selectivity can be important for studies aiming to test the Maxwell-Boltzmann velocity distribution for confined gases“. I am not sure I understand this. How would one select all different velocity components?

8. Methods: Gas, not Gaz

9. Methods: Pressure broadening and shift: “In these experiments, the pressure gauge is naturally placed outside the thin-cell (see Fig. 2a of the main text) and a pressure differential cannot be excluded (cell thickness is $\sim 5\mu\text{m}$), contrary to selective reflection measurements [30], performed with the same pressure gauge (and the same set-up) but in a macroscopic cell“. Too long and hard to read. Better: “In these experiments, the pressure gauge is naturally placed outside the thin-cell (see Fig. 2a of the main text). Thus, a pressure differential cannot be excluded (cell thickness is $\sim 5\mu\text{m}$), contrary to selective reflection measurements [30], performed with the same pressure gauge (and the same set-up) but in a macroscopic cell“.

10. Methods: Pressure broadening and shift: “Nevertheless, both techniques yield very similar pressure broadenings suggesting that gas pressure readings outside the cell, also reflect the pressure inside the cell.” I would remove the comma after “cell” and perhaps add one after “broadenings“.

Reply to referees

We would like to thank all the referees for their constructive comments that have allowed us to improve our manuscript. All referees seem to agree on the importance of our results, the quality of measurements compared to the challenges that we faced in this experiment, and the quality of the manuscript. Here, we reply to the referee's comments and we highlight (in red color) the corresponding changes in the main text (and Methods).

Reviewer 1:

- 1) *There is no mention to an absolute frequency determination and the quality of the calibration of the frequency axis seems to be quite poor (1% in relative term) as compared to the stat of the art. Moreover, in consideration of a possible application of the method for frequency metrology, I believe that some extra efforts should be done in this direction. Why do not to use a frequency comb as reference (at least in the 1.5 micron region)?*

For an absolute calibration of the frequency scale we use a saturated absorption reference whose FWHM linewidth is typically ~ 1 MHz. Therefore, the absolute frequency scale is known with a precision better than the MHz, given that most transition frequencies (with the exception of SF₆ hotbands in the 10.6 μ m range) are tabulated with a precision comparable, (for most transitions tabulated in the HITRAN database) or better, (in the case of acetylene) than the MHz. The frequency scan calibration is then performed with a wavemeter or by using multiple tabulated transitions as references.

The above method has been repeatedly used by our group to analyze the spectra of confined atoms of linewidths ~ 10 MHz [see for example A. Laliotis et al. Nat Commun. 2014 where the frequency scale calibration was also corroborated by beating the scanning laser with a second source stabilized by saturated absorption spectroscopy]. Although some systematic errors can occur in the case of very long scans, frequency calibration has not been a major source of uncertainty in our measurements. Moreover, similar calibration methods are also typically used by molecular spectroscopists to pinpoint multiple molecular rovibrations in the near and mid-infrared. Using a frequency comb would be in some sense the ultimate way of calibrating our frequency scale but we believe that it is not necessary for the purpose of the present experiments that aim at exploring the physics of confined molecules (as shown for atoms, a comb is not necessary for Casimir-Polder measurements) and at providing a novel way of performing rovibrational molecular spectroscopy. Here we provide frequencies with an accuracy on the order of MHz (sufficient for molecular databases) with the particular advantage of measuring also transition *amplitudes* (due to the linearity of the method in optical power). A frequency comb could be necessary at a later stage if thin-cells prove well adapted for ultra-high precision (below 100 kHz) spectroscopy. Regarding frequency metrology applications, our aim is to develop stand-alone thin-cell acetylene based frequency references. An additional absolute standard (such as a referenced comb) will then only be necessary for characterizing the performance of the thin-cell based reference.

Changes in the text: We add some more information on absolute frequency calibration on the Methods section.

- 2) What about uncertainties? What about possible systematics effects? An overall uncertainty budget has been completely omitted. From the "statistical" point of view, it is surprising for me that the observed pressure broadening coefficients are given without uncertainty. Are they derived from repeated measurements? Are they experimental results? How can the authors claim consistency with literature values?

The pressure broadening coefficients are experimentally measured. They are extracted by measuring the homogeneous linewidth (Γ) at various pressures. Our data suggests a linear increase of Γ with pressure that allowed us to measure the pressure broadening coefficients with a statistical accuracy better than 10% (depending on the transition). One source of systematic error comes from the model used to reproduce thin-cell spectra (see our answer to point 3). However, in our opinion the largest systematic error, affecting the measurements reported in the initial version of the manuscript, came from pressure measurements that can suffer from gauge calibration, pressure fluctuations and, in the case of confined molecular gases, pressure differentials (inside/outside the cell). We indicatively mention that the acetylene pressure broadenings reported in the literature can vary between 8-13MHz/Torr for the same P9 transition.

In this new version of the text we calibrate our pressure measurements using a capacitive gauge. This measurement corrects the systematic errors related to pressure calibration and allows us to confidently report pressure broadenings and shifts with reduced uncertainties (below 10%).

Changes in the text: We summarize our results in the new version of Table 1 including also a discussion in the main body of the text. A detailed description of the measurements is given in the Methods section.

- 3) *Results. From a quantitative point of view, the experiment provides results that are limited to pressure broadening coefficients (of one transition of each investigate molecule). This is not too much. Maybe, It is enough for a proof of principle experiment, but not for the initial ambitions of the work.*

Here we demonstrate a method for fabricating thin cells and perform high-resolution thin-cell spectroscopy of confined molecules. In our opinion, the most important *quantitative* results are the observed signal *amplitude* and *lineshape* of the probed molecular gases and the excellent agreement of the experimental spectra with our theoretical model. The pressure broadening of the probed molecular rovibrations is an additional result, relevant for applications in molecular spectroscopy.

Probing subwavelength confined molecular gases is an important breakthrough that raises fundamental questions on the interaction and collisions of quantum particles (atoms or molecules) with surfaces. The remarkable agreement between theory and experiment suggests that our results are compatible with the assumptions included in the model, concerning the effects of collisions on laser-particle interaction, on the velocity distribution and on the partition function of desorbed molecules. Some of these assumptions have been questioned in the literature of confined atomic vapours (see for example [52,53,56]) while experiments with molecular gases are scarce [28]. Molecules can offer specific advantages compared to atoms, allowing us to probe rovibrations (transitions within the same electronic state) with high resolution and test the limits of the assumptions included in the theoretical models. This could allow us to test the effects of collisions on the rotational partition function (redistribution of the rotational quantum number, J , after collision) and perform strenuous tests of the Maxwell-Boltzmann distribution.

We agree, however, with the referee that the measured pressure broadenings (Γ) and shifts (δ) should be highlighted in the text demonstrating the potential of thin-cells for molecular spectroscopy. We therefore amend the manuscript accordingly (see also our answer to point 2). We stress that our measurements of Γ , δ are obtained with a linear, high-resolution spectroscopic method allowing sub-Doppler resolution signals at pressures typically inaccessible with saturated

absorption (efficient at low pressures) or Doppler absorption (efficient at very high pressures where Γ compares to the Doppler width).

Changes in the text: We added a paragraph in the discussion section highlighting the important quantitative results of our work, consisting in the observed signal *amplitude* and *lineshape* and agreement with theory. We also added a few lines in the main text better explaining the assumptions of the model. Finally, we amended the manuscript to highlight the measured pressure broadenings (Γ) and shifts (δ).

- 4) *Laser sources used in the experiment. It seems to me that they represent the predominant limitation for the experiment (at least in the MIR. In the case of acetylene, pressure broadening conceals this limitation). This is disappointing. I am sure that the authors (see Optica 6 411-423, 2019) could have access to a "more powerful" and SI traceable QC laser, not limited to a linewidth at the MHz level. Again, as a proof of principle experiment this is not a problem, but when the publication targets for NC, extra efforts have to be done.*

The set-up used in Optica 6 411-423 (2019) aims at precision frequency measurements of methanol rovibrations with a precision better than 10kHz. This is performed with a QCL stabilized to a frequency comb that is stabilized on an ultra-stable reference. Such ultra-high precision spectroscopy is beyond the scope of this work, and the possibility of using thin-cell platforms in this domain is in itself an ambitious project (due to signal to noise considerations) that cannot be tackled in this paper. The breakthrough reported here is probing and studying subwavelength confined molecules. Our work has indeed applications in molecular spectroscopy but these are oriented towards pinpointing unknown molecular transition *amplitudes* and *frequencies* (suitable for molecular databases) as well as measuring pressure broadening and shifts with sub-Doppler resolution at pressures inaccessible by saturated absorption spectroscopy. In this context the current laser linewidth is acceptable.

Indeed, for some applications in molecular spectroscopy or more importantly in Casimir-Polder spectroscopy reducing the laser linewidth below 100kHz could be a clear advantage. For such dedicated experiments, we need to identify the most suitable molecule and subsequently use the technical know-how of our team to reduce the laser linewidth.

- 5) *The authors have in their hands a very powerful method. Indeed, the hyperfine structure of ammonia has been observed. Why do not provide evidence of consistency (at least in terms of detuning of the different components) with previous experimental results or theoretical data?*

We follow up on the referee's suggestion and present studies on the hyperfine structure of ammonia. In particular, we study the global collisional shift and we test for differential collisional shifts between the hyperfine components checking for consistency with other experiments at low pressures. We believe that the observed global collisional shift is a new result that strengthens this paper. At low pressures the hyperfine splitting is consistent with reported values (see reference [51]) to within 0.1MHz or better. This was verified using different ways of calibrating the frequency scale of our spectra (as mentioned in the Methods section).

Changes in the text: We include these results in the main text.

- 6) *"Discussion" paragraph. Usually, it should be a physical insight related to the results of the experiment described in the manuscript. In this work, it is a description of what can be done in future, as prospective. It is absolutely not quantitative*

According to the suggestion of the referee, we add a paragraph in our discussion section in order to stress the novelty of our results and discuss the implications for the physics of confined gases. We also give some more quantitative information on the aspects of increasing the amplitude of our signal by a means of a Fabry-Perot cavity (also suggested by referee 2 and 3). We nevertheless believe that the discussion section should qualitatively present some perspectives of the reported work and cannot quantitatively cover all the possible challenges faced in future experiments.

Changes in the text: We added a paragraph in the discussion section giving some insight on the importance of our quantitative results. This new addition has prompted us to change the order of some paragraphs in order to improve the flow of the discussion section. We also add some quantitative information concerning the finesse of the FP resonator (last paragraph of the text).

- 7) *“Thin-cell fabrication” paragraph: More appropriate for a technical report (or for the Method section).*

Thin-cell fabrication is one of the results of our paper. We therefore believe that this short paragraph gives the reader a full picture of the novelty of our experiment. We give more extensive details in the methods section.

Reviewer 2:

- 1) *It would be useful if it would be mentioned what the Q-factor of such a cavity would be. Is the currently achieved alignment sufficient for effective enhancement?*

Indeed, we believe that placing the cell within a Fabry-Perot cavity could be a route towards increasing the signal amplitude. In the Methods section we provide the signal amplitude as a function of the reflection coefficient, for a symmetric cavity of $\lambda/2$ thickness (this is done under the approximation of a homogeneous linewidth (Γ) much smaller than the Doppler width, which is valid in the case of acetylene spectroscopy). The equation (proportional to the finesse of the cavity for high values of the reflection coefficient) suggests that in order to achieve an amplification of 100 one needs a reflectivity of $R \sim 0.98$ and therefore a finesse of 150-200. This would require a control of the cell thickness on the order of a few nanometers over an area corresponding to the beam size. For a beam size of $\sim 100 \mu\text{m}$ the average thickness gradient of $\sim 100 \text{nm/cm}$ (or locally better, achieved in the thin-cell fabricated for this work) is more than sufficient. To facilitate alignment, we can (although we do not mention it in this paper) locally fine tune the cell thickness (at the point of the beam) by applying external pressure with an annular piezoelectric element. This could also allow tuning the cavity resonance at multiple acetylene rovibrations. An external Fabry-Perot cavity (also suggested in our paper) would pose no challenges in terms of alignment. Here, the maximum achievable finesse would be limited by the AR coating on the windows (to minimize reflection losses). However, finesse values of several hundred should be achievable.

Changes in the text: We try to give some more quantitative elements on this subject (also according to referee 1 suggestion) in what is now the last paragraph of our text.

- 2) *Also it would be helpful to mention how the particular transitions were picked – I assume they are simple the one closest to the wavelength that fits the cell!?*

We fabricated a cell with thickness corresponding to $\sim \lambda/2$ to probe molecular transitions around $10.6 \mu\text{m}$ that can be reached with our QCL laser (emitting from $10.53\text{-}10.58 \mu\text{m}$). In this range, our laser can probe the NH_3 P(1,0) transition (the only strong NH_3 transition in this frequency range) chosen as a good candidate for probing Casimir-Polder interactions (see a discussion in [30]) and

v_3 rovibrations of SF₆. Here, we chose to present a typical SF₆ spectrum (close to the NH₃ line) but in reality multiple v_3 SF₆ transitions are accessible with our set-up (essentially limited by our laser). We plan to present a list of SF₆ rovibrations (hot-bands) in a future paper dedicated to molecular spectroscopy, presenting the compatibility of our data with molecular databases. We would like to stress that our thin-cell could also be used to probe other molecules in the fingerprint region, where rovibrations are typically strong (C₂H₄ at 10.5 μ m, CH₃OH at 9.7 μ m...), even if half the transition wavelength ($\lambda/2$) does not exactly fit the cell thickness. The P9 transition of acetylene at 1530nm was chosen because it corresponds to $\sim 7\lambda/2$ in the uniform and comfortable area of our cell. It is also one of the strongest acetylene combination lines at telecom wavelengths. We try to make these points clearer in the text.

Changes in the text: We try to make these points clearer in the text, explain why the specific transitions were chosen.

Reviewer 3:

- 1) *In the introduction, the authors correctly cite the relevant literature and note that thin-cell molecular spectroscopy has been elusive until now. I find difficult to guess why the experiments presented here have not been done years ago, for all necessary ingredients seem to be available now for quite some time. I would appreciate if the authors could make an effort to rationalise such delay. What are the key points that allowed the present measurements? As a reader I would find this interesting.*

We have tried to give an explanation for this time delay in the introduction of our manuscript. The major difficulty with molecules was, in our opinion, the unavailability of easy-to-use sources in the mid infra-red (such as laser diodes for the near infra-red used to probe atomic excitations), where molecular transitions are typically strongest. QCL sources have bridged that gap but it also took a significant time (after initial commercialization) to render QCL sources compatible for high resolution spectroscopy. We mention that some of our experiments performed with CO₂ lasers at 10.6 μ m showed that these sources are hard to adapt to the needs of selective reflection spectroscopy (limited scan range, difficulties with frequency modulation...). Apart from the above 'objective' factors explaining the delay between atomic and molecular thin cells, we also believe that our experiment presents several advances (automated pressure on/off modulation technique, QCL frequency control, Fabrication of a thin-cell with ZnSe windows...) that made the reported breakthrough of molecular thin-cells possible.

- 2) *Caption to Fig 1: I would repeat at the text beginning that the data are calculations (is stated only in the main text).*
OK
- 3) *Caption to Fig 1: I find this sentence strange: "This oscillatory, coherent, behavior eventually disappears for larger L (when the collisional mean free path becomes smaller than the cell thickness) and a sub-Doppler peak is observed for all thicknesses". First they say that the oscillatory behaviour eventually disappears and then they say that is observed for all thicknesses. I would reformulate this and the previous sentences as: "The sub-Doppler contribution observed for $L=\lambda/2$ is revived when $L=(2n+1)\lambda$ (coherent Dicke-narrowing). It is observed up to $7\lambda/2$ and disappears for larger L (when the collisional mean free path becomes smaller than the cell thickness)". Or something similar.*

We modify the phrase to 'This oscillatory behavior (collapse and revival of the sub-Doppler peak) eventually disappears...' in order to stress that it is the collapse and revival (oscillatory behavior) that disappears and not the sub-Doppler peak. For large thicknesses a sub-Doppler peak is always observed. It is however, overshadowed by the linearly growing Doppler broadened contribution.

- 4) *Caption to Fig 1: Since they give an idea of the vertical scale, would it be possible to write a non-arbitrary scale to the vertical axes? Perhaps in terms of some relevant parameters used in the simulations?*

The vertical scale of Fig1a will depend on the dipole moment matrix element of the given molecular transition. This can vary by more than an order of magnitude between NH_3 and C_2H_2 for instance. Using an arbitrary vertical axis allows Fig1a to apply to all molecular transitions. We would therefore prefer to maintain it.

- 5) *In the discussion "Gaining a few orders of magnitude in signal can allow stabilization of a telecommunications laser...". Can the authors give a more precise estimate of the required gain and quantify the expected gain from the proposed improvements? How far away are those applications?*

We have tried to be more quantitative in the new version of the text. This is an ongoing project in our group. An important step towards this application is to externally fine-tune the alignment of the cavity using pressure by an annular piezoelectric element (see our reply to reviewer 2).

- 6) *Discussion: Something is missing from the sentence: "A global analysis of high resolution SF_6 spectra (such as the one performed in [54]), including thin-cell spectroscopy experiments could potentially offer unprecedented information on hot-band transition amplitudes and positions for SF_6 and other heavy atmospheric molecules...". Perhaps an initial "In"?*

We have rewritten the above sentence as: *Including thin-cell spectroscopic data in a global analysis of SF_6 spectra (such as the one performed in [62]) could potentially offer unprecedented information on hot-band transition amplitudes and positions for SF_6 and other heavy atmospheric molecules (ClONO_2 , CF_4) in the future.*

- 7) *Discussion: "Furthermore, the flexibility in temperature operation, coupled with high-frequency resolution and high velocity selectivity can be important for studies aiming to test the Maxwell-Boltzmann velocity distribution for confined gases". I am not sure I understand this. How would one select all different velocity components?*

The velocity class resonant with the laser beam (Δu) is roughly proportional to the ratio Γ/Δ_D (homogenous linewidth over Doppler width). Higher frequency resolution means therefore better velocity selection (smaller Δu) which could be an advantage for testing the velocity distribution of molecules departing from the surface. In the new version of the text we have removed this phrase that can perturb the reader and is difficult to explain succinctly. The possibility of testing the Maxwell-Boltzmann distribution with confined molecules is mentioned in the new paragraph added in the conclusions.

- 8) *Methods: Gas, not Gaz*
OK.

- 9) *Methods: Pressure broadening and shift: "In these experiments, the pressure gauge is naturally placed outside the thin-cell (see Fig. 2a of the main text) and a pressure differential cannot be excluded (cell thickness is $\sim 5\mu\text{m}$), contrary to selective reflection measurements [30], performed with the same pressure gauge (and the same set-up) but in a macroscopic cell". Too long and hard to read. Better: "In these experiments, the pressure gauge is naturally placed outside the thin-cell (see Fig. 2a of the main text). Thus, a pressure differential cannot be excluded (cell thickness is $\sim 5\mu\text{m}$), contrary to selective reflection measurements [30], performed with the same pressure gauge (and the same set-up) but in a macroscopic cell".*

OK

- 10) *Methods: Pressure broadening and shift: "Nevertheless, both techniques yield very similar pressure broadenings suggesting that gas pressure readings outside the cell, also reflect the pressure inside the cell." I would remove the comma after "cell" and perhaps add one after "broadenings".*

OK

REVIEWERS' COMMENTS

Reviewer #1 (Remarks to the Author):

The revised version of the manuscript looks much better than the original paper. It seems to me that all the comments raised by the three referees have been properly addressed.

I appreciate the work the authors have done in revising the manuscript and, therefore, in my opinion, it can be accepted for publication in NC.

Reviewer #2 (Remarks to the Author):

I am happy with the responses to my remarks and have no further comments. In my opinion, the manuscript should be published without further delay.

Reviewer #3 (Remarks to the Author):

I think that the manuscript is good for publication in the present form. The authors have addressed all relevant issues, either by improving the manuscript or by giving good answers to the raised questions.